# Characterisation of Cartilage Damage via Fusing Mid-Infrared, Near-Infrared, and Raman Spectroscopic Data

**DOI:** 10.3390/jpm13071036

**Published:** 2023-06-24

**Authors:** Rubina Shaikh, Valeria Tafintseva, Ervin Nippolainen, Vesa Virtanen, Johanne Solheim, Boris Zimmermann, Simo Saarakkala, Juha Töyräs, Achim Kohler, Isaac O. Afara

**Affiliations:** 1Department of Technical Physics, University of Eastern Finland, 70211 Kuopio, Finland; ervin.nippolainen@uef.fi (E.N.); juha.toyras@uef.fi (J.T.); isaac.afara@uef.fi (I.O.A.); 2School of Physics, Clinical and Optometric Sciences, Technological University Dublin, D07 XT95 Dublin, Ireland; 3Faculty of Science and Technology, Norwegian University of Life Sciences, 1432 Ås, Norway; valeria.tafintseva@nmbu.no (V.T.); johanne.heitmann.solheim@nmbu.no (J.S.); boris.zimmermann@nmbu.no (B.Z.); achim.kohler@nmbu.no (A.K.); 4Research Unit of Medical Imaging, Physics and Technology, University of Oulu, 90570 Oulu, Finland; simo.saarakkala@oulu.fi (S.S.); vesa.k.virtanen@oulu.fi (V.V.); 5Research Unit of Health Sciences and Technology, University of Oulu, 90220 Oulu, Finland; 6Science Service Center, Kuopio University Hospital, 70210 Kuopio, Finland; 7School of Information Technology and Electrical Engineering, The University of Queensland, Brisban, QLD 4072, Australia

**Keywords:** post-traumatic osteoarthritis, articular cartilage, data fusion, partial least squares discriminant analysis

## Abstract

Mid-infrared spectroscopy (MIR), near-infrared spectroscopy (NIR), and Raman spectroscopy are all well-established analytical techniques in biomedical applications. Since they provide complementary chemical information, we aimed to determine whether combining them amplifies their strengths and mitigates their weaknesses. This study investigates the feasibility of the fusion of MIR, NIR, and Raman spectroscopic data for characterising articular cartilage integrity. Osteochondral specimens from bovine patellae were subjected to mechanical and enzymatic damage, and then MIR, NIR, and Raman data were acquired from the damaged and control specimens. We assessed the capacity of individual spectroscopic methods to classify the samples into damage or control groups using Partial Least Squares Discriminant Analysis (PLS-DA). Multi-block PLS-DA was carried out to assess the potential of data fusion by combining the dataset by applying two-block (MIR and NIR, MIR and Raman, NIR and Raman) and three-block approaches (MIR, NIR, and Raman). The results of the one-block models show a higher classification accuracy for NIR (93%) and MIR (92%) than for Raman (76%) spectroscopy. In contrast, we observed the highest classification efficiency of 94% and 93% for the two-block (MIR and NIR) and three-block models, respectively. The detailed correlative analysis of the spectral features contributing to the discrimination in the three-block models adds considerably more insight into the molecular origin of cartilage damage.

## 1. Introduction

Globally, over 250 million people are affected by osteoarthritis (OA) [1]. The worldwide prevalence and socioeconomic impacts of osteoarthritis are influenced by population growth, ageing, and obesity [2]. Moreover, it has been estimated that there is a delay of nine years in diagnosing OA. OA has been projected to become one of the most prevalent diseases in developed countries in the upcoming decade [1]. In Europe, the annual cost of OA ranges from EUR 1330 to EUR 10,452 per person [3].

OA is often attributable to ageing (idiopathic) or trauma (post-traumatic) and is associated with severe pain, restricted joint mobility, and articular cartilage matrix erosion. At present, there is no cure for OA; however, several repair strategies are available during arthroscopy to treat joint injuries and prevent OA development and progression. The efficacy of a repair technique fundamentally depends on an accurate diagnosis of the severity of the lesion [4]. Usually, during arthroscopic surgery, the surgeon’s treatment decision depends on visual evaluations and manual probing of the cartilage tissue [5]. However, this method is highly subjective and has poor reproducibility [6,7]. Consequently, research efforts are focused on developing objective, sensitive, and reproducible tools. Over the past few years, there has been a growing interest in utilising biophotonic (the interplay between optics and biological material) methods to evaluate connective tissue health [8,9,10,11].

Vibrational spectroscopic methods such as near-infrared (NIR), mid-infrared (MIR), and Raman spectroscopies are emerging techniques in biomedical applications and have demonstrated significant diagnostic potential for evaluating cartilage properties [8,10,12]. However, these techniques have their strengths and weaknesses; for instance, as light penetration in biological tissues is wavelength-dependent, MIR radiation in attenuated total reflectance (ATR) measurement only penetrates a few microns into tissues compared to 1–5 mm for NIR and Raman spectroscopies [13]. Hence, MIR spectroscopy makes the superficial cartilage highly molecularly sensitive, which may provide information related to tissue degeneration in early OA. In contrast to MIR spectroscopy, NIR and Raman spectroscopies can be applied to monitor OA changes in deep cartilage tissue and subchondral bone. Moreover, it is well-known that Raman scattering is sensitive to non-polar molecular groups (such as C-C, C=C). At the same time, MIR absorption is sensitive to polar molecular groups (such as C-O, C=O).

In comparison, a NIR spectrum captures the features from absorption arising from the interactions between light and chemical bonds such as OH, CH, NH, and CO in cartilage. Thus, they may provide complementary information regarding tissue integrity [14]. It is possible to harness their strengths while compensating for their weaknesses by combining them in a multi-modal approach using data fusion [15] and obtaining complementary information.

Data fusion or data blocking refers to combining two or more datasets from different modalities to obtain greater insights than one would if one used the datasets individually [14]. Data fusion is a powerful tool used to generate models that are more reliable than individual modalities [14,16,17]. This approach is widely used in numerous areas, including robotics, metabolomics, image processing, intelligent system design, plant science and food analysis [16,17,18,19]. However, it has not been sufficiently explored in biomedical optics and could be valuable for understanding the progression of complex diseases such as OA.

This study investigates the potential of data fusion/data blocking of MIR, NIR, and Raman spectroscopic data for characterising degenerative changes in articular cartilage. We utilise previously reported datasets [20,21,22] on the three spectroscopic modalities to determine if their combination yields more benefits than an individual spectroscopic modality would. We hypothesised that data fusion offers an additional advantage over individual spectroscopic methods in terms of detecting cartilage degeneration. To test this hypothesis, bovine osteochondral specimens were subjected to mechanical and enzymatic damage, and MIR, NIR, and Raman spectra were acquired from the samples. The capacity of NIR, MIR, and Raman spectroscopy to differentiate damaged cartilage and healthy cartilage was measured using Partial Least Squares Discriminant Analysis (PLS-DA) via one-block (individual spectroscopic modalities) and multi-block (different combinations of the modalities) datasets.

## 2. Materials and Methods

### 2.1. Sample Preparation

Sample extraction: Fresh bovine knees (n = 12, age 60–95 weeks) were obtained from a local abattoir; no ethical permission was needed. A total of 12 patellae were collected. Among them, two patellae showed signs of natural damage; hence, they were excluded from the study. Samples were extracted from the patellae and subjected to different damage treatments within one hour of patellae extraction from the joints. The cartilage surfaces of the patellae were kept hydrated during this process by placing PBS-soaked gauze on the cartilage surfaces. Specimens were kept hydrated between the measurements by immersing them in PBS. All the measurements were carried out from the centre of the same sample.

Only ten patellae with no visible signs of damage were extracted from the knee joints and utilised for experiments. Cylindrical osteochondral specimens (n = 60, diameter = 7 mm) were removed from different anatomical locations of the medial and lateral sides of the patellae (Figure 1I) and prepared for artificial damage to mimic physiological injuries and degeneration. Based on the tissue damage procedure, specimens were divided into mechanical damage (M) (n = 24) and enzymatic damage (E) (n = 36) groups. Respective control samples (n = 60) for each test specimen were extracted from adjacent tissues on the patellae (Figure 1I).

Mechanical damage: The mechanical damage group was further divided into two sub-groups: Impact damage (M1, n = 12) and abrasion (M2, n = 12). Specimens in group M1 were subjected to mechanical damage using a custom-made drop tower (Figure 1IIa). In brief, impact damage was induced by dropping a stainless-steel ball (*m* = 200 g, diameter = 1 cm) from a height of 7.5 cm. The drop heights of the impactor and the energy delivered to the cartilage surface were determined based on preliminary testing to produce chondral cracks. The impactor was removed from the specimens immediately after the impact damage to avoid creep deformation. Samples were placed in phosphate-buffered saline (PBS) and allowed to recover for one hour. Specimens in group M2 were subjected to mechanical abrasion to fibrillate the surface of cartilage via a custom-made grinding device (Figure 1IIb). The cartilage surface was abraded by applying constant stress of 4 kPa on a rotating (180°) metal plate covered with sandpaper (particle size 200 µm). The protocol was repeated in two perpendicular grinding directions. Samples were then allowed to recover for one hour in PBS.

Enzymatic damage: The enzymatic damage group was divided into three sub-groups based on the enzyme type and incubation time (Figure 1IIc). Two enzymes were utilised for cartilage degradation: collagenase and trypsin. Collagenase D (0.1 mg/mL, Sigma Aldrich) was used to disrupt the collagen network, while trypsin (0.5 mg/mL, T4299, Sigma-Aldrich Inc., St. Louis, MO, USA) was used to digest the proteoglycan macromolecules (with minor collateral effect on the collagen network) [20,21,22]. Two different incubation times (90 min and 24 h) were used for collagenase D treatment.

The three enzymatic treatment subcategories were collagenase 24 h (E1 n = 12), collagenase 90 min (E2, n = 12), and trypsin 30 min (E3, n = 12). Before degradation, the samples were pre-warmed to 37 °C. They were then incubated at 37 °C and 5% CO_2_ in PBS containing the respective enzymes and complementary antibiotics (Sigma-Aldrich Inc., St. Louis, MO, USA). The incubation time of 24 h for collagenase induced severe cartilage damage, thus simulating advanced tissue degeneration. In contrast, incubation times of 30 min for trypsin and 90 min for collagenase were applied to induce mild damage, simulating early-stage cartilage degeneration. The enzymatic treatments were conducted on larger rectangular tissues samples (10 × 9 × 15 mm) to minimise the impact of lateral penetration of enzymes into the cartilage from the cut edges, followed by extraction of cylindrical plugs (d = 7 mm) from the centre of the samples. All specimens were used in the experiments within 10 h of extraction from the knee joints. Samples were kept in PBS before and after measurements to keep them hydrated.

For further analysis, the different damaged cartilage sub-groups were pooled into the “damaged cartilage” group, and the corresponding control samples were pooled into the “normal cartilage” group. Thus, the classification problem in this study was to differentiate between damaged and healthy samples.

### 2.2. Instrumentation and Spectroscopic Measurements

NIR Spectroscopy: NIR spectra were acquired using a spectrometer (AvaSpecULS2048XL, Avantes BW, grating 75 lines/mm, slit 50 lm, which gives in k = 1.0–2.5 lm resolution = 6.4 nm), a light source (Avalight-HAL-S, Avantes BW, Netherlands), and a custom-made diffuse reflectance arthroscopic fibre-optic probe. The stainless-steel fibre-optic probe had a tip (d = 3.25 mm) shaped like a conventional arthroscopic hook. The probe consisted of 114 optical fibres, with 100 fibres illuminating the sample from the light source and 14 fibres collecting the diffuse reflected light from the sample to the spectrometer.

NIR spectra were obtained from the exact location before (pre) and after (post) degradation, and the samples were immersed in PBS during measurement to imitate in vivo arthroscopy. Three spectra were acquired per sample from the centre of the specimen using Avasoft software (version 8.7.0, Avantes BV). Each spectral acquisition consisted of 50 co-added scans, with an integration time of 16 ms.

Raman Spectroscopy: Raman spectra were acquired using a DXR2xi Raman microscope (Thermo Fisher Scientific, Madison, WI, USA). The configuration and parameters of our experiments were as follows: laser wavelength 785 nm, laser power of 30 mW, 200–3400 cm^−1^ full-range grating, 10X objective, and 50 μm confocal pinhole. The spectral integration time for each Raman spectrum was 0.5 s, and 120 scans per spectrum were averaged. The measurements were carried out using the OMNICxi software. Experimental parameters were selected based on preliminary experiments to maximise the signal-to-noise ratio and minimise sample auto-fluorescence. We adopted the 785 nm laser as it is optimal for this application. It has been observed that 785 nm laser excitation can largely avoid fluorescence compared to 532 nm, which can swamp the Raman signals with sample background fluorescence.

Specimens were placed on the sample stage, and Raman spectra (3 spectra per sample) were acquired from the centre of each sample. To keep the samples hydrated, they were stored in PBS before and after taking measurements. The total measurement time for all three spectra per specimen was under 5 min.

Mid-infrared Spectroscopy: MIR spectra were recorded using a Thermo Nicolet iS50 FTIR spectrometer (Thermo Nicolet Corporation, Madison, WI, USA). The system consisted of a global MIR source and a liquid nitrogen-cooled mercury cadmium telluride (MCT) detector and was coupled with a custom-made ATR probe (Art Photonics GmbH, Berlin, Germany).

Samples were kept moist and placed on the stage, and three MIR spectra were recorded per sample from the centre of each sample. MIR spectra were acquired by first establishing contact between the sample and the ATR crystal of the probe. Spectral data were then recorded using the following acquisition parameters: spectral resolution 2 cm^−1^, digital spacing 0.2411 cm^−1^, averaging 64 scans, and spectral range 4000 cm^−1^–400 cm^−1^. Prior to sample measurement, a background spectrum was recorded from the air. All measurements were carried out using the OMNIC software (Thermo Nicolet Corporation, Madison, WI, USA).

### 2.3. Data Analysis

Spectral data from the three modalities (NIR, MIR and Raman spectroscopy) were analysed using the following data analysis workflow: (i) data pre-processing, (ii) building one-block classification models, (iii) building multi-block classification models. The models differentiating between damaged and healthy samples were developed using the PLS-DA algorithm, and classification accuracy was evaluated using a leave-one joint-out cross-validation approach, i.e., in each step of cross-validation, samples from one bovine knee were taken out entirely from the training set and utilised as an independent test set to assess the model’s performance.

Data pre-processing: Before data analysis, all raw spectral data (NIR, Raman, and MIR) were pre-processed to reduce spectral data variance originating from sources such as light scattering, background signals, or instrumental artefacts. Spectral pre-processing was conducted using nippy (https://github.com/uef-bbc/nippy (accessed on 23 June 2023)), an open-source pre-processing toolbox [23] developed in-house. Nippy contains various pre-processing options and can be adapted with customised functions. Pre-processing methods from five categories were utilised, such as clipping, standard normal variate (SNV), derivation, smoothing, and trimming. The parameters of the pre-processing approaches were optimised based on the classification accuracy of the established models. The exact pre-processing parameters used in the study are provided in Table 1.

The optimal pre-processing methods were different for each spectroscopic dataset (NIR, MIR, and Raman), a highlighted in Table 1. This is expected considering the differences in underlying phenomena, e.g., absorption and scattering, and how these are influenced by physical effects, such as the structural makeup of samples.

One-block models: The pre-processed data were subjected to PLS-DA analysis, and classification models were built using one data block at a time. PLS-DA is a multivariate classification method that regresses matrix Y (response variables) onto matrix X (predictor variables), where Y is a matrix of class belongingness, i.e., 0 (normal cartilage) or 1 (damaged cartilage) values, while X is a matrix of spectral data. The models were optimised using leave-one joint-out cross-validation, and the number of latent variables (LVs) were selected to minimise the prediction error of cross-validation. Details of LVs are provided in their respective tables.

Multi-block models: Multi-block analysis was carried out in a two-block fashion by combining two spectroscopic data blocks at a time (MIR and NIR; MIR and Raman; NIR and Raman data) and in a three-block style by combining all three data blocks (MIR, NIR, and Raman). The data corresponding to different techniques (MIR, NIR, and Raman) were used in the multi-block PLS-DA analysis [19]. Multi-block analysis requires sample-to-sample correspondence between blocks of data. One sample was excluded from all blocks in our datasets since it was removed from the Raman dataset due to poor spectral quality. Each X block was normalised by its Frobenius norm, i.e., the square root of the sum of squared values of the matrix elements. Such scaling was performed to set the different data blocks on the same footing and avoid the weighting of blocks due to differences in the numeric values between, e.g., MIR absorbances and Raman scattering intensities. The same approach for model optimisation was used for the one-block analysis.

## 3. Results and Discussion

MIR, NIR, and Raman spectroscopy are non-destructive, rapid, label-free, and complementary techniques that provide information on articular cartilage structure and molecular composition [10,21,22,24]. This study is the first to assess data fusion as a potential methodology for characterising articular cartilage integrity from spectroscopic data. Essentially, this study evaluates the efficiency of the data fusion approach to identify damaged cartilage using MIR, NIR, and Raman spectroscopic datasets as opposed to each individual (one-block models) dataset.

The classification accuracy of the one-block PLS-DA models was highest for NIR data (with a sensitivity of 90% and a specificity of 97%), followed by MIR data (sensitivity and specificity of 92%) and then Raman (sensitivity = 83% and specificity = 68%) data (Table 2a). The three-block model showed a relative improvement in performance, with a sensitivity of 92% and specificity of 93% (Table 2b). The sensitivity and specificity of the classification two-block models improved for all data fusion options (Table 3), with the combination of MIR and NIR datasets yielding the best synergistic effect with a sensitivity of 92% and specificity of 95%. The sensitivity and specificity of one-block and multi-block models are presented in Figure 2 [25].

The scatter plot and block weights for a three-block model combining MIR, NIR, and Raman to differentiate between the healthy and damaged groups are presented in Figure 2. Multi-block models have three levels of interpretation; the first two ways can consider the global level represented by the scatter plots and block weight (Figure 2a,b).

The influence of the weights for each data block (MIR, NIR, and Raman) on the three-block model are shown in Figure 2b. The MIR block is observably important in both the 1st and 2nd LVs, and Raman is observably strongly related to the 1st LV and NIR to the 2nd LV (Figure 2b). Some of the NIR score plot patterns are visible in the 2nd LV of the global score plot, while some of the Raman block patterns are visible in the 1st LV of the global score plot.

The third way to interpret the three-block model is by using Regression coefficients—the actual regression model—which allows for interpretations at the lowest level. By inspecting the regression coefficients, one can observe the differences between samples in different groups and identify influential spectral bands that differentiate samples. The information obtained from the regression coefficients of each block was used to create the correlation loading plot presented in Figure 3.

The correlation loading plot (Figure 4) shows the relationship between the most critical variables of the different data blocks and design parameters, i.e., the treatment groups. As the bands with the highest regression coefficients and representing the most discriminative bands of the spectral data, the variables for the correlation loading plot were pre-selected based on the regression coefficients of the three-block model.

The peaks between the blue and red circles are interconnected by a correlation coefficient between 0.5 (at the dotted red circle line) and 1 (at the blue circle line) (Figure 4). The correlation approaches zero when moving to the centre of the plot. The points closer to the centre are not well explained by LV1 and LV2; therefore, correlations among the variables close to the centre may need to be explored in other LVs.

The MIR spectra features (highlighted in blue) provide the most important variables: strong positive and negative correlation among the peaks of amides I, II, and III bands, and the ester peak at 1740 cm^−1^ is correlated to some peaks of the amide bands (Figure 4). The MIR spectral peaks also exhibit a correlation with the Raman peaks (highlighted in orange) in the first LV. The most critical variables of the NIR spectra (highlighted in red) can be observed along the LV2, which shows minimal correlation with a majority of the MIR and Raman peaks. NIR peaks, such as those at 4726 and 4829 cm^−1^, appear to be correlated with the amide peaks of MIR spectra at 1655 and 1564 cm^−1^. This suggests that NIR spectroscopy provides some complementary information to the other blocks in the model.

We also observed that the model somewhat better explains the variation due to E1 (collagenase 24 h treated) group. Therefore, this group correlates with some of the peaks in the 1st LV more than the other damage groups (E2 and 3, M1 and 2).

Although the results show a relatively low classification accuracy for the one-block model based on the Raman data compared to the NIR and MIR data, we believe this could be due to different sampling volumes; for example, MIR and NIR spectra were acquired using fibre-optic systems, whereas Raman spectra were obtained using a microscopic system. The spot sizes of the MIR, NIR, and Raman systems were 2.5 mm, 2 mm, and 3.8 μm, respectively, resulting in different sampling volumes. Moreover, as multiple chondral (cartilage) cracks were observed in the impact damage group, sampling with a smaller spot size might result in higher variation.

The multi-block (two-block and three-block) models showed similar or improved performance compared to the one-block models. The improvement between the two-block (MIR and NIR) model (94% accuracy) and one-block NIR model (93% accuracy) can be considered relatively marginal, and the same can be said for the three-block (MIR, NIR, and Raman; 93% accuracy) and one-block NIR models. However, in terms of sensitivity and specificity, the models built on multi-block data, such as the two-block or three-block models, appeared to be more stable than the one-block models.

The most significant advantage of multi-block models lies in their interpretability rather than increased classification accuracy. Therefore, a vital advantage of the multi-block models was to provide an overview of the combined datasets and knowledge of how each block, represented by the different spectroscopic techniques (separately and combined), contributes to the discrimination between normal and damaged samples.

Although advanced methods, such as neural networks and deep learning, have advantages over classical machine learning methods, they require a large amount of data—much larger than what was used in this study—to develop robust models. The deep learning methods applied for spectroscopic data did not perform much better than the classical machine learning methods in classification models [19,26]. Augmenting such data to improve the classification via the use of deep learning methods is not a straightforward task. While NIR spectroscopy excels when the goal of the analysis is simple tissue classification, the combination of MIR and NIR spectroscopy can provide an added advantage in terms of interpretation. For example, the two-block (MIR and NIR) model can enable the estimation of the relative contribution of superficial (from MIR) and deep (from NIR) cartilage features to differentiate between healthy and damaged tissue. Furthermore, another key strength of the multi-block model lies in its potential for interpreting and evaluating the contribution of different tissue constituents and biochemical species to the classification models.

The NIR and Raman datasets provide additional discriminatory information since the model based on MIR data only is worse in differentiating the groups compared to the three-block model (Table 2 and Table 3, Figure 2a). The correlation loading plots (Figure 3) provide a visualisation of the relationship between the peaks of MIR, NIR, and Raman spectra that are responsible for the three-block PLS-DA model.

Several studies have explored the implementation of multi-modal data fusion to predict blood pressure, stroke or to identify abdominal cancer [27,28,29]. Different data fusion approaches have been described in the literature [28,30,31]; some are based on data compression by PCA and PLS methods before modelling [32,33]. However, we did not utilise such approaches in this paper because such systems require large datasets where an independent test set can validate the model performance. Optimising the data compression and implementing the discrimination model would require more test datasets. Furthermore, unsupervised techniques for data compression are preferable to supervised methods when no external validation is possible. Moreover, data compression makes the model interpretation difficult, resulting in a “black-box” model, which is difficult to interpret and evaluate.

## 4. Conclusions

We implemented multi-modal data fusion methods, going beyond the black-box techniques and providing further insight into the spectral feature-based models. The results show that analysing MIR, NIR, and Raman spectra through the use of multi-block models accurately identifies experimental cartilage damage. Multi-block analysis has the added value of enabling more detailed interpretations compared to the one-block models, exhibiting the improved multi-modal models attributed to the biochemical changes in cartilage tissue. The information reported in this study could help develop multi-modal (two or three blocks) spectroscopic techniques to determine the optimal surgical treatment during cartilage repair via arthroscopy.

## Figures and Tables

**Figure 1 jpm-13-01036-f001:**
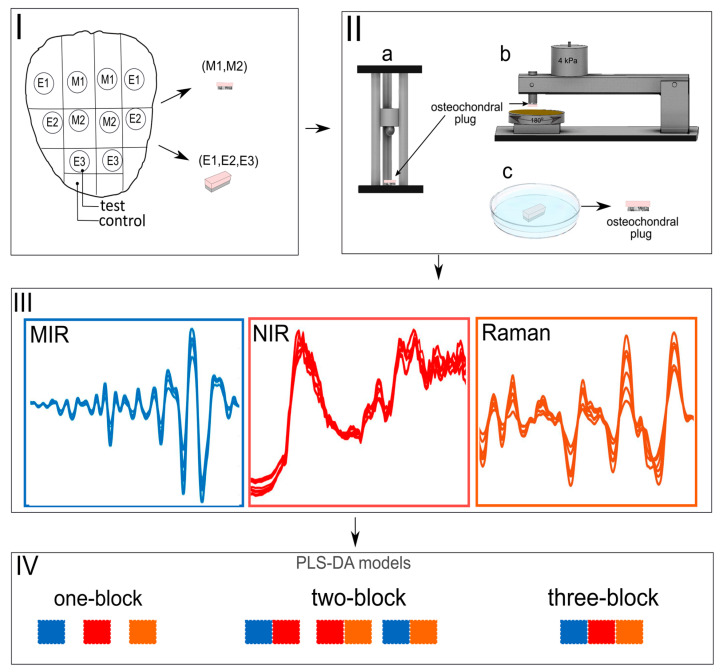
Study Protocol, (**I**) Anatomical location of osteochondral samples extracted from bovine patella: M1-impact damage, M2-abrasion, E1-collagenase 24 h, E2-Collagenase 90 min, E3-trypsin (**II**) (a) custom-made drop-tower used to create impact injury on cartilage (**III**) Mid-infrared (MIR), Near-infrared (NIR) and Raman spectral data (**IV**) one-block and multi-block PLS-DA models.

**Figure 2 jpm-13-01036-f002:**
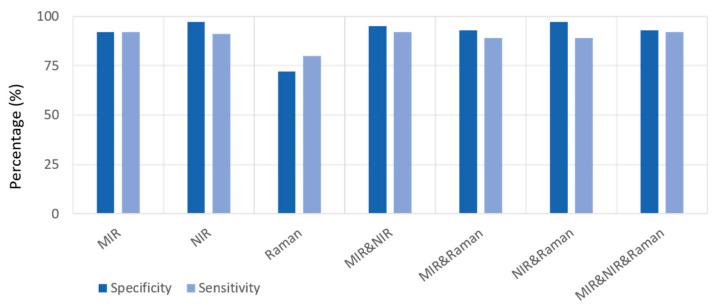
Sensitivity and Specificity of one-block and multi-block PLS-DA models.

**Figure 3 jpm-13-01036-f003:**
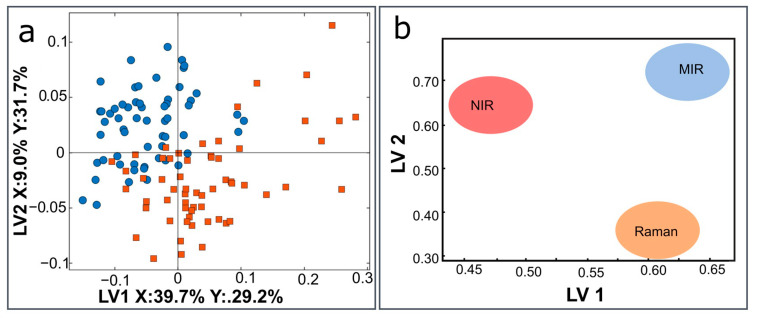
Three-block classification model-damaged vs. healthy groups (**a**) Scatter plot, (**b**) block weights plot for MIR (mid-infrared), NIR (near-infrared), and Raman data on the PLS-DA model.

**Figure 4 jpm-13-01036-f004:**
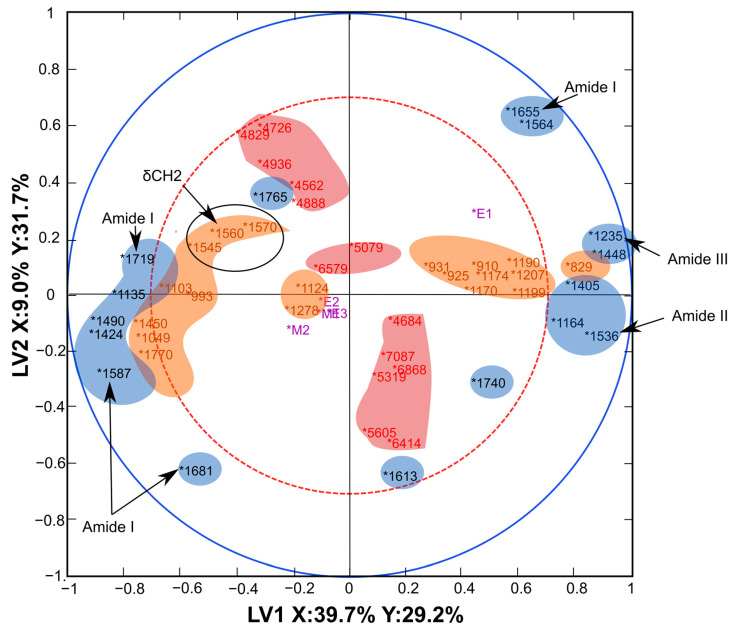
Correlation loading plot: MIR peaks (cm^−1^) in blue, NIR peaks (cm^−1^) in red, Raman peaks (cm^−1^) in orange, treatment groups in purple (E1: collagenase 24 h, E2: collagenase 90 min, E3: trypsin 30 min, M1-impact damage, M2-abrasion).

**Table 1 jpm-13-01036-t001:** Pre-processing parameters for Mid-infrared (MIR), Near-infrared (NIR), and Raman spectra.

Method	Pre-Processing Parameters		
	Derivative	SNV	Trimming (cm^−1^)
MIR	order-2, filter window-201, polynomial order-3	yes	900–1800
NIR	order-0, filter window-5, polynomial order-3	yes	8333–4545
Raman	order-1, filter window-205, polynomial order-3	yes	800–1800

**Table 2 jpm-13-01036-t002:** Classification accuracies of (a) one-block and (b) three-block models.

(a) One-Block Models, Classification Accuracy (%)		
**Mid-infrared, accuracy = 92%, LV * = 10**		
	Damage	Healthy
Damage (n = 60)	**92**%	8%
Healthy (n = 60)	8%	**92**%
**Near-infrared, accuracy = 93%, LV * = 5**		
	Damage	Healthy
Damage (n = 60)	**90**%	10%
Healthy (n = 60)	3%	**97**%
**Raman, accuracy = 76%, LV * = 4**		
	Damage	Healthy
Damage (n = 59)	**83**%	17%
Healthy (n = 60)	32%	**68**%
**(b) Three-block model**		
*Mid-infrared, Near-infrared and Raman, accuracy = 93%, LV * = 5*		
	Damage	Healthy
Damage (n = 59)	**92**%	8 %
Healthy (n = 60)	7%	**93**%

* LV = Latent Variables.

**Table 3 jpm-13-01036-t003:** Two-block models: (a) Mid-infrared (MIR) and Near-infrared (NIR). (b) Mid-infrared (MIR) and Raman. (c) Near-infrared (NIR) and Raman.

(a) MIR and NIR, Accuracy = 94%, LV * = 3		
	Damage	Healthy
Damage (n = 59)	**92**%	8%
Healthy (n = 60)	5%	**95**%
**(b) MIR and Raman, Accuracy = 91%, LV * = 8**		
	Damage	Healthy
Damage (n = 59)	**88**%	12%
Healthy (n = 60)	7%	**93**%
**(c) NIR and Raman, Accuracy = 92%, LV * = 7**		
	Damage	Healthy
Damage (n = 59)	**88**%	12%
Healthy (n = 60)	3%	**97**%

* LV = Latent Variables.

## Data Availability

Data can be made available upon reasonable request to the authors.

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
