# Peer review of "Characterisation of Cartilage Damage via Fusing Mid-Infrared, Near-Infrared, and Raman Spectroscopic Data"

_jpm, 2023, doi:10.3390/jpm13071036_

Round 1

Reviewer 1 Report

In my opinion, the main issue is for the authors to demonstrate a statistical metric that demonstrates whether there are significant differences in the classification rates of the 1-block and multi-block models. This is very important because behind this metric there are several analytical issues behind it, such as analysis time, instrumental accessibility, analytical frequency, among others.

Author Response

Reviewer 1:

In my opinion, the main issue is for the authors to demonstrate a statistical metric that demonstrates whether there are significant differences in the classification rates of the 1-block and multi-block models. This is very important because behind this metric are several analytical issues, such as analysis time, instrumental accessibility, and analytical frequency, among others.

Response: We are grateful to the Reviewer for pointing this out. We do not expect a statistical difference between models, as the values are very close to each other; another issue is that we have three models each for one-block and two-block models, and only one model of a three-block model, which creates technical difficulty in carrying out such calculations. Moreover, the manuscript explains that Classification models should not be evaluated solely based on classification accuracy but should go beyond the black-box techniques.

Also, the second Reviewer raised this issue, highlighting that classification accuracy could not be the only metric to interpret the results. Hence, we have calculated the sensitivity and specificity of one-block and multi-block models.

Action: Page 7, Line 257- In the revised manuscript, we added Figure-2 with sensitivity and specificity of one-block and multi-block models and the following text in the results and discussion section.

Figure 2 Sensitivity and Specificity of one-block and multiblock PLS-DA models

Reviewer 2 Report

The article entitled "Characterisation of Cartilage Damage via Mid-Infrared, Near-Infrared and Raman Spectroscopic Data Fusion" is an engaging work on spectroscopy and multivariate analysis application for cartilage damage detection and classification. The Authors aimed to classify samples and assess the usefulness of different spectroscopy techniques (NIR, MIR and Raman) and their combination in characterising degenerative changes in articular cartilage. 

The Introduction provides the appropriate background, explaining the pros and cons of spectroscopy techniques in analysing biological samples. The Authors also wonder whether aggregating these techniques in various combinations may allow them to obtain a broader spectrum of information on the tested materials. 

In the next, Material and Methods section, the Authors describe in detail all the procedures used to obtain analysed articular cartilage samples and their mechanical and enzymatic damage methodology. PLS-DA models planned to be used are also clearly presented and explained, as well as the procedures of sample measurements using MIR, NIR and Raman spectroscopy.

The Results and Discussion section presents obtained classification records and the evaluation of spectroscopic data obtained using the three tested methods and their combinations in terms of their usefulness for assessing cartilage condition. The Authors refer to the results obtained by other researchers and explain the limitations of their research.

In my opinion, the article is prepared carefully, has a high scientific potential and is suitable for publication with minor corrections, which I present below:

- how were the samples for MIR analysis prepared (Dried? If yes, how long was the drying time and which method was used?)

- The assessment of the classifier using only accuracy might need to be clarified, as accuracy is not the best metric for evaluating how a model performs. The Authors should consider more metrics to describe the classifier in a broader range (https://www.nature.com/articles/nmeth.3945)

The quality of the language used is correct, requiring only minor adjustments

Author Response

Reviewer 2:

The article entitled "Characterisation of Cartilage Damage via Mid-Infrared, Near-Infrared and Raman Spectroscopic Data Fusion" is an engaging work on spectroscopy and multivariate analysis application for cartilage damage detection and classification. The Authors aimed to classify samples and assess the usefulness of different spectroscopy techniques (NIR, MIR and Raman) and their combination in characterising degenerative changes in articular cartilage. 

The Introduction provides the appropriate background, explaining the pros and cons of spectroscopy techniques in analysing biological samples. The Authors also wonder whether aggregating these techniques in various combinations may allow them to obtain a broader spectrum of information on the tested materials. 

In the next, Material and Methods section, the Authors describe in detail all the procedures used to obtain analysed articular cartilage samples and their mechanical and enzymatic damage methodology. PLS-DA models planned to be used are also clearly presented and explained, as well as the procedures of sample measurements using MIR, NIR and Raman spectroscopy.

The Results and Discussion section presents obtained classification records and the evaluation of spectroscopic data obtained using the three tested methods and their combinations in terms of their usefulness for assessing cartilage condition. The Authors refer to the results obtained by other researchers and explain the limitations of their research.

In my opinion, the article is prepared carefully, has a high scientific potential and is suitable for publication with minor corrections, which I present below:

- how were the samples for MIR analysis prepared (Dried? If yes, how long was the drying time and which method was used?)

Response: We thank the Reviewer for the encouraging and positive comments. We measured the samples using a mid-infrared probe, and the samples were kept moist before measuring the MIR data. This was done as we wanted to maintain samples as naturally as possible. This was deemed important as this study is part of a larger EU- funded MIRACLE project in which we developed a mid-infrared fibre-optic system (https://miracleproject.eu/).

Action: Page 5, Line -184, To improve clarity, we modified the Materials and Methods section line: "Samples were kept moist and placed on the stage…”

- The assessment of the classifier using only accuracy might need to be clarified, as accuracy is not the best metric for evaluating how a model performs. The Authors should consider more metrics to describe the classifier in a broader range (https://www.nature.com/articles/nmeth.3945)

Response: We thank you very much for this insightful comment. We agree with the Reviewer that the accuracy does not provide a complete picture of the model’s performance. This is primarily for the highly imbalanced data where one class can be much more significant in terms of samples and thus overrepresent the classification performance. The suggested article nicely summarises the other metrics that should be used in addition to the accuracy and the effect of the class imbalance on the selected metrics. The other examples of the metrics are True Positive rate (TP), True Negative rate (TN), False Positive rate (FP) and False Negative rate (FN). However, the class imbalance is not an issue in our study since the two classes, healthy and damaged cartilage, are nicely balanced.

For our models, we provide so-called confusion matrices described in the reference provided by the Reviewer. Such confusion matrices contain all the metrics mentioned above, i.e. TP, TN, FP and FN. The confusion matrix for the binary classification, which is the case of our models, is provided on one diagonal, TP and TN, while FP and FN metrics are on the other. The higher the TP and TN, the better, while FP and FN should be small. This provides the idea of the skewness in the classification towards one of the classes and the general performance of the model. Our models show overall good performance as both TP and TN are high and quite similar, while FN and FP are pretty low. The only exception is probably the one-block Raman model that shows a bit skewed performance towards the damaged class: TP or sensitivity = 83% and TN or specificity = 68%.

Action: In the revised manuscript, we have added Figure 2, depicting sensitivity and specificity across all the models, and added the following lines explaining the observations.

We have also calculated the sensitivity and specificity of one-block and multi-block models (Figure 2) [26]. Page 7, Line 257

Figure 2 Sensitivity and Specificity of one-block and multiblock PLS-DA models

The quality of the language used is correct, requiring only minor adjustments.

Response: We thank the Reviewer for this positive comment; we have improved the text during the revision process.